# InsPro: Propagating Instance Query and Proposal for Online Video Instance Segmentation

**Fei He**[1,2], **Haoyang Zhang**[4], **Naiyu Gao**[4], **Jian Jia**[1,2], **Yanhu Shan**[4],
**Xin Zhao**[1,2*], **Kaiqi Huang**[1,2,3]

[1] CRISE, Institute of Automation, Chinese Academy of Sciences
[2] School of Artificial Intelligence, University of Chinese Academy of Sciences
[3] CAS Center for Excellence in Brain Science and Intelligence Technology  [4] Horizon Robotics

{hefei2018,jiajian2018}@ia.ac.cn, {haoyang.zhang,naiyu01.gao,yanhu.shan}@horizon.ai
{xzhao,kaiqi.huang}@nlpr.ia.ac.cn

## Abstract

Video instance segmentation (VIS) aims at segmenting and tracking objects in videos. Prior methods typically generate frame-level or clip-level object instances first and then associate them by either additional tracking heads or complex instance matching algorithms. This explicit instance association approach increases system complexity and fails to fully exploit temporal cues in videos. In this paper, we design a simple, fast and yet effective query-based framework for online VIS. Relying on an instance query and proposal propagation mechanism with several specially developed components, this framework can perform accurate instance association implicitly. Specifically, we generate frame-level object instances based on a set of instance query-proposal pairs propagated from previous frames. This instance query-proposal pair is learned to bind with one specific object across frames through conscientiously developed strategies. When using such a pair to predict an object instance on the current frame, not only the generated instance is automatically associated with its precursors on previous frames, but the model gets a good prior for predicting the same object. In this way, we naturally achieve implicit instance association in parallel with segmentation and elegantly take advantage of temporal clues in videos. To show the effectiveness of our method InsPro, we evaluate it on two popular VIS benchmarks, *i.e.*, YouTube-VIS 2019 and YouTube-VIS 2021. Without bells-and-whistles, our InsPro with ResNet-50 backbone achieves 43.2 AP and 37.6 AP on these two benchmarks respectively, outperforming all other online VIS methods.

## 1 Introduction

Video instance segmentation (VIS) [1] is a challenging but important computer vision task. It requires not only segmenting object instances on each video frame but also associating them across all frames. Due to its fine-grained object representation form, it has got a wide range of applications in various areas such as autonomous driving and video editing.

Existing VIS methods can be categorized into two groups: frame-level methods and clip-level methods. Frame-level methods [1, 2, 3, 4] generally follow a 'tracking-by-detection' paradigm, which first generate per-frame object instances by existing instance segmentation models [5, 6], and then associate them across frames via additional tracking heads (as shown in Figure 1 (a)). In comparison, clip-level methods [7, 8, 9, 10] take a 'clip-matching' paradigm, which divide a video

---

*Corresponding author

36th Conference on Neural Information Processing Systems (NeurIPS 2022).

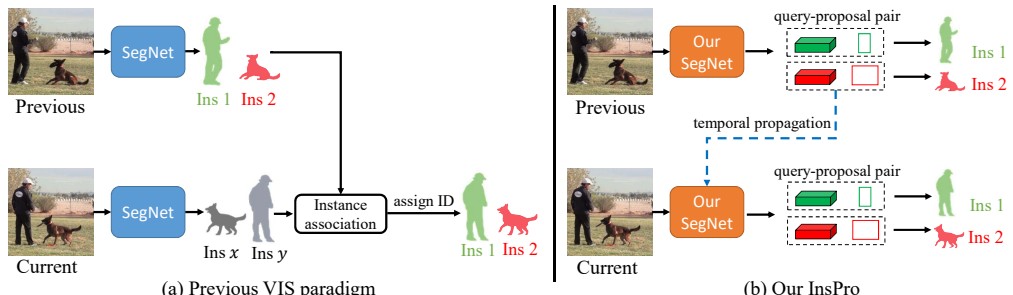

| | |
|---|---|
| (a) Previous VIS paradigm | (b) Our InsPro |

Figure 1: (a) Previous methods take a two-step approach to VIS. They first generate object instances and then perform explicit instance association to link them across frames. (b) Our InsPro implements implicit instance association through a temporal propagation mechanism, achieving object instance segmentation and tracking in one shot. It generates frame-level object instances based on a set of instance query and proposal pairs propagated from previous frames. Since the instance query-proposal pair is learned to represent one specific object across frames, the instance association is naturally achieved in parallel with segmentation and video temporal clues are elegantly exploited meanwhile.

into multiple overlapped clips, generate instance predictions for each clip, and then associate these clip-level predictions by some hand-crafted instance matching algorithms. Whether frame-level or clip-level methods, both of them inevitably need an explicit instance association step to fulfill object tracking. This generally requires to design a complicated association strategy to achieve good tracking performance, which is not trivial. More importantly, the explicit association step increases model complexity and slows inference speed. Furthermore, this extra step also indicates that the temporal clues intrinsic in videos are not well utilized, as the instance prediction is performed separately on each frame or each clip.

In this work, inspired by the recent success of query-based object detectors [11, 12], we propose a simple, fast and yet effective query-based framework for online VIS. Our system, dubbed as InsPro, segments and tracks objects in one shot through an instance query and proposal propagation strategy with carefully designed modules (Figure 1 (b)) , which eliminates the explicit instance association step. Specifically, our approach generates frame-level object instances based on a set of instance query-proposal pairs propagated from previous frames. In the learning process, we develop several techniques to make sure that the generated instance query-proposal pair corresponds to one specific object across frames. Thus, when an object instance is generated using such a query-proposal pair on the current frame, it is automatically associated with its precursors on all previous frames. In this way, we achieve implicit object association without a linking step. Meanwhile, this instance query-proposal propagation mechanism also enables our VIS system to achieve a better prediction accuracy (see Table 1). This benefits from the instance query-proposal pair's encoding one object's temporal and spatial information across all previous frames, which provides a very good prior for the model to infer the same object on the current frame. In this sense, our query-based VIS method actually implements an efficient way to exploit the intrinsic temporal clues in videos.

To fulfill the advantages of our VIS system, learning exclusive and expressive instance query-proposal pairs is the key. In this work, we develop several strategies to ensure the learning effectiveness. First, we design a temporally consistent matching mechanism to enforce the one-to-one correspondence between the instance query-proposal pair and a specific ground truth object across frames during training. Second, we propose a box deduplication loss to enlarge the distance between instance proposals. This helps suppress duplicate proposals on the same object and increase the exclusivity of the generated instance query-proposal pair. At the same time, the sparsely distributed unoccupied query-proposal pairs can serve as candidates in the next frame to detect new objects, allowing our system to achieve new object detection and tracking effortlessly. Third, we propose an intra-query attention module that enhances instance query with its predecessors encoding the same object. This explicitly aggregates long-range object information into the query, augmenting its representation capacity, which helps handle occlusion and motion blur.

To validate the effectiveness and efficiency of our InsPro, we conduct extensive experiments on two popular VIS benchmarks [1], *i.e.*, YouTube-VIS 2019 and YouTube-VIS 2021. Without bells-and-whistles, our InsPro with ResNet-50 [13] backbone achieves 43.2 AP on YouTube-VIS 2019 and 37.6 AP on YouTube-VIS 2021 respectively, outperforming all other online VIS models. Moreover, our

lite variant, InsPro-lite, reaches 38.7 AP on YouTube-VIS 2019 at impressive 45.7 FPS on a Nvidia RTX2080Ti GPU.

In summary, we make the following contributions in this paper. 1) We propose a simple, fast and yet effective query-based framework for online VIS. 2) We develop several techniques to make the query-proposal pair propagation mechanism work smoothly. These techniques distinguish our work from other query propagation-based object association methods [14, 15, 16], and make our work simpler, more elegant and more effective than them. 3) Our VIS system achieves the state-of-the-art performances on two popular VIS benchmarks.

## 2 Related Work

**Frame-level VIS Methods**   mainly adopt a 'tracking-by-detection' paradigm and can run in an online fashion. They first generate instance predictions frame by frame and then perform explicit instance association. MaskTrack R-CNN [1] first proposes the VIS task and simply adds an additional tracking head to Mask R-CNN [5] for instance association. Follow-up works [2, 3, 4, 17] improve either the segmentation or the tracking algorithm to achieve better performance. On the other hand, some works [8, 18, 19, 20, 21, 22] attempt to perform temporal feature fusion to improve instance segmentation and association. For example, PCAN [22] proposes frame- and instance-level prototypical cross-attention modules to leverage rich spatio-temporal information to facilitate better segmentation. All these methods require additional modules to achieve explicit instance association, which expands model complexity and reduces inference speed. By contrast, our method performs instance association implicitly through an instance query and proposal propagation mechanism, which is simpler and naturally exploits the temporal and spatial consistency in videos.

**Clip-level VIS Methods**   take a 'clip-matching' paradigm, which process multiple frames within a clip simultaneously and then perform instance matching between clips to complete VIS. While some methods [7, 23, 9] propagate instance information within a clip with well-designed propagation modules to model temporal context, recent works [8, 10] utilize transformer [24] to model temporal context in an end-to-end manner. These methods normally need hand-crafted matching algorithms to complete instance association between clips. Although they usually achieve high performance, they can only run in an offline mode, which restricts their application to limited areas. In contrast, our method achieves comparable performance but can run online.

**Query-based Methods**   have attracted increasing attention in recent years due to their flexibility and simplicity. DETR [11] first uses a set of learned queries interacting with image features to encode objects, and then directly outputs detections by decoding the transformed queries. Following works [25, 26, 27, 28, 29] improve DETR in terms of either training efficiency or detection performance. Sparse R-CNN [12] builds a query-based detector on top of R-CNN architecture [30, 31]. Its follow-up works [3, 32] extend it to instance segmentation and video object detection. Besides, Max-DeepLab [33] proposes a box-free panoptic segmentation method with external query. DAFL [34] uses a set of queries to encode pedestrian information for pedestrian attribute recognition.

The success of DETR has also inspired query-based VIS methods. VisTR [8] adapts DETR to the VIS task. It takes a video clip as input and directly outputs the sequence of masks for each instance orderly. IFC [10] proposes inter-frame communication transformers to reduce the heavy computation and memory usage of VisTR-like VIS methods. Similar to VisTR, Mask2Former [35] applies masked attention to a video clip and directly predicts a 3D instance volume. To learn a powerful video-level instance query, SeqFormer [36] aggregates temporal information from each frame to the instance query. These methods work on clips rather than frames, and achieve object association through sharing of the queries within a clip rather than query propagation. Thus, they still need instance matching between clips. Instead, our method applies to frames, and can propagate query-proposal pairs through the entire video and thereby can associate object instances over any frame length.

**Query Propagation-based Object Association Methods**   have been recently explored in several works, such as TransTrack [14], TrackFormer [15], MOTR [16] and EfficientVIS [37], which are also inspired by query-based methods [11, 12]. This shows the effectiveness and potential of such a new object linking approach. The differences between our InsPro and them are as follows.

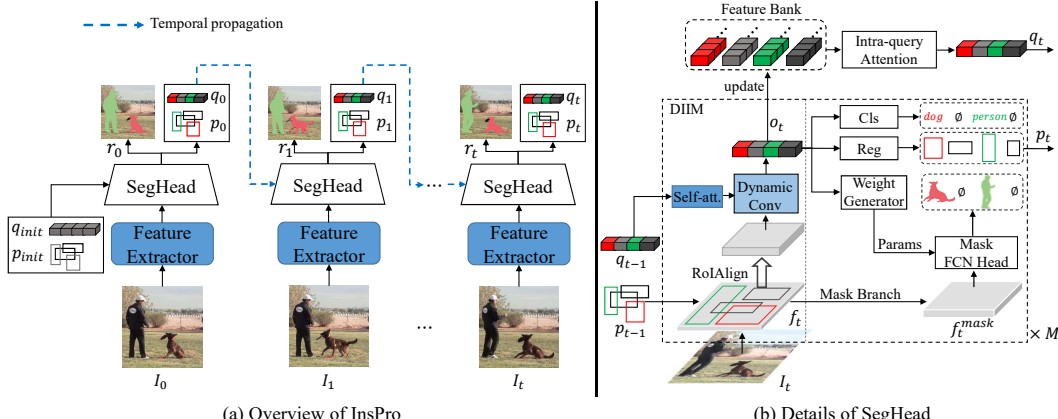

(a) Overview of InsPro                    (b) Details of SegHead

Figure 2: (a) Overview of our InsPro. It performs VIS by propagating instance query-proposal pairs across frames. $q_{init} \in \mathbb{R}^{N \times C}$ and $p_{init} \in \mathbb{R}^{N \times 4}$ are initial instance queries and proposals on the first video frame, respectively. They are used in SegHead to predict instance results $r_0$ on frame $I_0$, and to produce updated $q_0$ and $p_0$ which are propagated to the next frame. By repeating this process, we complete the VIS task. (b) Details of SegHead. It is a multi-stage network, consisting of a dynamic instance interaction module (DIIM) and an instance segmentation module. The former transforms instance queries with RoI features of corresponding proposals and produces object features, while the latter predicts object instances based on the object features and conditional convolution [39].

First, our InsPro is different from them in the way of either tracking seen objects or detecting new objects. TransTrack is basically a 'tracking-by-detection' method, because it still needs to explicitly match detection boxes to tracked boxes in each frame, while our InsPro performs implicit association. More importantly, TransTrack, TrackFormer and MOTR adopt a track query subset to track seen objects and an extra object query subset to detect new objects. This requires additional heuristic rules to combine two type queries, and may miss occluded or blurred objects with low scores, which can result in object trajectory break [38]. Our InsPro simply propagates all object queries produced in the previous frame to the current frame, and keeps using this set to track seen objects and detect new objects, which is much simpler and more elegant. As for EfficientVIS, our concurrent work, it does not consider this new object detection problem, and its performance will probably be impacted greatly if there are new objects in the next clip.

Furthermore, we design a more intelligent strategy to suppress duplicates. TransTrack and Track-Former employ score filtering or NMS to reduce duplicate predictions. MOTR builds a temporal aggregation network to learn more discriminative features to address this problem, while EfficientVIS does not discuss this problem. By contrast, we design a Box Deduplication Loss to suppress duplicates and an Inter-query attention module to enhance queries with their predecessors. Our solution avoids heuristic rules and post-processing steps, and is more effective according to the experimental results (see Table 2 (e)).

## 3 InsPro

We aim to design a simple and fast VIS system that performs instance association implicitly and exploits video temporal clues elegantly. To this end, we take a query-based VIS approach that predicts object instances on each frame based on a set of instance query-proposal pairs propagated from previous frames. In this section, we introduce our VIS system, InsPro, including an instance query and proposal propagation mechanism and an instance segmentation head. Meanwhile, we also describe those proposed techniques that make our propagation mechanism work well.

### 3.1 Instance Query and Proposal Propagation

The instance query and proposal propagation mechanism enables our VIS system to perform object instance association implicitly in parallel with instance segmentation. Since it is inspired by the recent query-based object detector Sparse R-CNN [12], we first briefly review Sparse R-CNN.

Sparse R-CNN [12] formulates object detection as a set prediction problem and achieves state-of-the-art performance. It simplifies the detection pipeline and removes heuristic components like NMS. Specifically, it first initializes a fixed set of learnable instance queries ($N \times C$, $N$ denotes the number of queries and $C$ the query dimension) paired with learnable instance proposals ($N \times 4$) to describe objects in an image. As illustrated in Figure 2 (b), each instance query is convolved with the RoI feature of the corresponding proposal to output a more discriminate feature $o_t$ [12]. After multi-stage iterative updating, the instance query encodes more accurate object appearance information while the proposal captures more precise location. Finally, decoding the object feature $o_t$ produced using the instance query-proposal pairs, we get the detection results.

Inspired by this instance query-proposal representation of an object, we design a query-proposal temporal propagation mechanism (as shown in Figure 2 (a)) to achieve implicit object instance association and temporal cue utilization in VIS. Our key insight is that there is a one-to-one correspondence between the learned instance query-proposal pair and a specific object. If we manage to preserve this correspondence from the first frame to the one where the object finally disappears, then we realize object tracking and object information propagation spontaneously.

To this end, we first initialize a set of instance queries $q_{init} \in \mathbb{R}^{N \times C}$ and proposals $p_{init} \in \mathbb{R}^{N \times 4}$ on the first video frame $I_0$, where $q_{init}$ and $p_{init}$ are learnable parameters and arranged in pairs. After learning, they are able to encode objects on the first frame. Decoding them with the first frame image feature inside the SegHead, we obtain instance results $r_0$ as well as a new set of updated pairs ($q_0$, $p_0$). Then we propagate this pair set ($q_0$, $p_0$) to the next frame as input to the SegHead. Similarly, we get the instance results $r_1$ and another new set of ($q_1$, $p_1$) on this frame. Among them, the object instance produced on this frame shares the same ID with the one on the previous frame *if they are both decoded by the same slice of the instance queries*. In this way, we automatically link object instances belonging to an identical object across frames and elegantly make use of object priors from the past. Repeating the above process until the last video frame, we then accomplish the VIS task on this video.

Please note that our InsPro simply propagates all object queries produced in the previous frame to the current frame, and keeps using this set to track seen objects and detect new objects. Instead, recent works [14, 15, 16] that take a similar query-propagation mechanism use a track query set to track seen objects and a new object query set to detect new objects respectively, which requires additional heuristic rules to combine these two type queries. Moreover, they rely on hand-crafted rules like a score threshold to select a subset of track queries, and occluded objects with low prediction scores are probably filtered out, which results in non-negligible true object missing and fragmented trajectories [38]. In comparison, our method is obviously simpler, more elegant and more effective (see Table 2 (e)).

**Intra-query Attention**   Since frame-by-frame temporal propagation encodes only short-range temporal cues, the instance query from just the last frame shows limitations in dealing with tough scenarios, *e.g.*, occlusion and motion blur. To boost the representation capacity of instance query, we augment it in practice with instance features from previous $T$ frames [40, 41]. Specifically, we build a feature bank that caches instance features from previous $T$ frames and perform intra-query attention inside this bank to aggregate long-range temporal cues into the current instance query, as shown in the upper part of Figure 2 (b). Formally, at frame $I_t$, instance features $o$ from previous $T$ frames are put together to form a feature bank $fb = \{o_{t-T+1}, \ldots, o_t\}$. Then, the enhanced instance query is computed as:

$$q_t^i = \frac{\sum_{n=0}^{T-1} o_{t-n}^i \exp(\varepsilon(o_{t-n}^i))}{\sum_{m=0}^{T-1} \exp(\varepsilon(o_{t-m}^i))} + o_t^i, \tag{1}$$

where $i$ denotes the $i$-th query and $\varepsilon(\cdot)$ is a linear transformation function. The enhanced $q_t$ is basically a weighted sum of instance features inside the feature bank, and the weights are learned upon the quality of the queries. Experiments (Table 6 (c)) show that this augmentation improves the query representation capacity greatly.

## 3.2   Segmentation Head

The segmentation head transforms the instance query and performs instance segmentation on each frame. As illustrated in Figure 2 (b), it is a multi-stage network and has two main parts: a dynamic instance interaction module and an instance segmentation module.

**Dynamic Instance Interaction Module** transforms the instance query with the proposal RoI feature and yields object features. It has $M$ stages and forms an iterative structure. At the first stage, given a pair of instance queries $\boldsymbol{q}_{t-1} \in \mathbb{R}^{N \times C}$ and proposals $\boldsymbol{p}_{t-1} \in \mathbb{R}^{N \times 4}$ that propagated from the previous frame $I_{t-1}$, it first augments the instance queries by a self-attention module [24], which models the inter-query relations. At the same time, it extracts the RoI feature of each proposal on the feature map by RoIAlign [5]. Then, each enhanced instance query convolves with its corresponding RoI feature through dynamic convolution [42] to get the object feature $\boldsymbol{o}_t \in \mathbb{R}^{N \times C}$. Since the object feature absorbs temporal cues encoded in instance queries, it has better representation ability than the single-frame RoI feature (see Table 2 (d)). The object feature is used to predict object instances in the following instance segmentation module, and the newly generated object boxes together with the object features proceed as input to the next stage in the iterative process. At the final stage, the object feature is processed together with its precursors from previous frames through the aforementioned intra-query attention module, to produce instance queries for the next frame.

**Instance Segmentation Module** decodes the object feature $\boldsymbol{o}_t$ and produces VIS predictions. It has three main heads. While the classification head predicts object classes, the regression one generates object boxes. Another mask head is responsible for producing instance masks through a conditional convolution [39] approach. Specifically, it first uses $\boldsymbol{o}_t$ to generate conditional convolution weights in weight generator. As shown in Figure 2 (b), it inputs $\boldsymbol{o}_t^i$ that represents the $i$-th object instance feature to the weight generator and outputs a set of convolution parameters $\boldsymbol{\omega}_i$. Meanwhile, it produces mask feature maps by transforming FPN [43] feature maps through a mask branch. Note that the output mask feature maps have 8 channels and a $\frac{1}{8}$ resolution of the input image, and are boosted by concatenating a 2-channel relative coordinates map to it. This relative coordinates map is computed using the center of predicted object boxes and provides strong location cues for predicting instance masks. Finally, we feed the combined feature maps $\boldsymbol{f}_t^{mask} \in \mathbb{R}^{10 \times \frac{H}{8} \times \frac{W}{8}}$ and convolution parameters $\boldsymbol{\omega}_i$ to a mask FCN head, predicting the $i$-th object instance mask via a conditional convolution:

$$\boldsymbol{m}_t^i = \text{CondConv}(\boldsymbol{f}_t^{mask}, \boldsymbol{\omega}_i). \tag{2}$$

For more details about the instance segmentation module, please refer to [39].

### 3.3 Temporally Consistent Matching

The key to the success of our InsPro is to make sure that the evolving instance query-proposal pair corresponds to the same object across frames in a video. To ensure this, one technique we propose is temporally consistent matching. This technique matches predictions and ground truth during training, assigns each ground truth object a proper prediction, and propagates the matching made on previous frames to subsequent frames.

Specifically, given a training batch consisting of multiple consecutive frames, we first compute the matching cost $L_{match}$ between predictions and ground truth objects on the first frame:

$$\mathcal{L}_{match} = \lambda_{cls} \cdot \mathcal{L}_{cls} + \lambda_{L1} \cdot \mathcal{L}_{L1} + \lambda_{giou} \cdot \mathcal{L}_{giou}, \tag{3}$$

where $\mathcal{L}_{cls}$ is the focal loss [44] between predicted classifications and ground-truth labels, $\mathcal{L}_{L1}$ and $\mathcal{L}_{giou}$ are L1 loss and the generalized IoU loss [45] between predicted boxes and ground-truth boxes, respectively. $\lambda_{cls}$, $\lambda_{L1}$ and $\lambda_{giou}$ are loss weights and set as 2, 5, and 2, respectively.

We search for the best bipartite matching that minimizes the matching cost $L_{match}$ with the Hungarian algorithm [46]. After finding the best matching on the first frame, we propagate this matching to other frames. Concretely, if one ground truth object still exists on subsequent frames, it will be matched to the prediction that is generated by the same instance query on the first frame. If there are new objects emerging, new matching will be made between the new objects and yet unmatched predictions. If a ground truth object disappears, its corresponding predictions will not participate in a new matching process. Through this temporally consistent matching mechanism, we bind one ground truth object to a single instance query during training.

### 3.4 Loss Function

**Box Deduplication Loss** Although the self-attention mechanism between queries has driven the model to generate fewer duplication predictions [11], we still observe multiple overlapped proposal

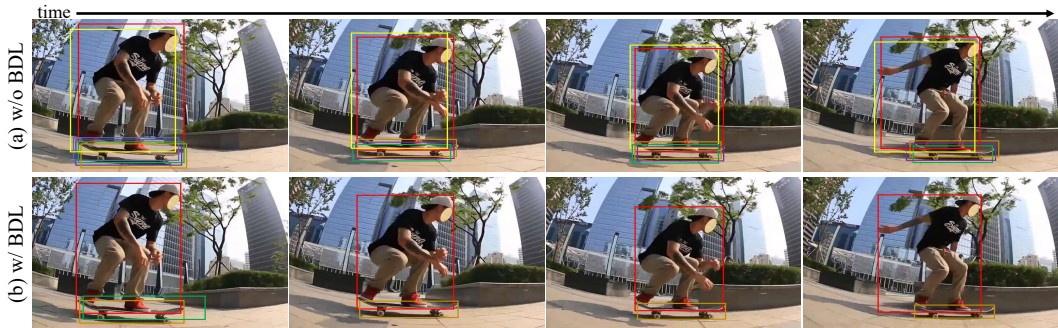

Figure 3: (a) Multiple duplicate boxes exist on the same object across frames. (b) After applying the proposed box deduplication loss (BDL) in training, the duplicate predictions are significantly suppressed along with temporal propagation.

boxes on the same object across many frames, as displayed in Figure 3 (a). We conjecture this is caused by those unmatched queries which cannot be pushed away from those matched queries due to lack of supervision. To address this problem, we propose a box deduplication loss to push away prediction boxes in terms of the center-to-center distance between them. As a result, not only the duplication problem is alleviated, but the sparsely distributed unmatched query-proposal pairs can serve as candidates in the next frame to detect and track new objects (see Figure 7 in Appendix). The loss is defined as:

$$\mathcal{L}_{dedup} = \frac{1}{k} \sum_{i=1}^{k} \max(\beta - \frac{\mathcal{C}^2(\boldsymbol{b}, \hat{\boldsymbol{b}}_{neg}^i)}{\mathcal{D}^2(\boldsymbol{b})}, \ 0), \tag{4}$$

where $\boldsymbol{b}$ is a ground truth box, $\hat{\boldsymbol{b}}_{neg}^i$ is a negative box that has the $i$-th largest IoU with $\boldsymbol{b}$ among those unmatched predicted boxes, $\mathcal{C}(\cdot)$ is the center distance calculation function, and $\mathcal{D}(\cdot)$ is the diagonal length calculation function. $\beta$ is set as 0.1 and $k$ as 5. This loss penalizes the short distance between $\boldsymbol{b}$ and $\hat{\boldsymbol{b}}_{neg}^i$, and drags all other duplicate boxes away from $\boldsymbol{b}$ [47]. With this new loss, our final box loss function is formed as:

$$\mathcal{L}_{box} = \lambda_{L1} \cdot \mathcal{L}_{L1} + \lambda_{\text{giou}} \cdot \mathcal{L}_{\text{giou}} + \lambda_{dedup} \cdot \mathcal{L}_{dedup}, \tag{5}$$

where $\lambda_{L1}$ and $\lambda_{\text{giou}}$ have the same values as in Equation 3, and $\lambda_{dedup}$ is set as 1.

**Overall Loss**  Given the one-to-one matching results, the final loss on each training frame is a sum of classification, box and mask losses:

$$\mathcal{L} = \lambda_{cls} \cdot \mathcal{L}_{cls} + \lambda_{box} \cdot \mathcal{L}_{box} + \lambda_{dice} \cdot \mathcal{L}_{dice} + \lambda_{focal} \cdot \mathcal{L}_{focal}, \tag{6}$$

where $\mathcal{L}_{dice}$ and $\mathcal{L}_{focal}$ are dice loss [48] and focal loss [44] for foreground mask prediction, respectively. We set $\lambda_{box} = 1$, $\lambda_{dice} = 5$ and $\lambda_{focal} = 5$. Finally, the losses of all training frames inside a batch are summed together and normalized by the number of frames.

## 4  Experiments

### 4.1  Datasets and Evaluation Metrics

We evaluate our method on YouTube-VIS 2019 and 2021 benchmarks [1]. YouTube-VIS 2019 consists of 2,238 training videos and 302 validation videos, and labels 40 object categories. YouTube-VIS 2021 is an extended version, which comprises 2,985 training videos and 421 validation videos, and labels improved 40 categories. All videos in these two datasets are annotated every 5 frames with object bounding box, object category, instance mask and instance ID. Following [1], we report the video-level average precision (AP) and average recall (AR) on the validation sets as the evaluation metrics, where both accurate instance segmentation and instance association are necessary to achieve high performance.

Table 1: Comparison of our InsPro to state-of-the-art methods. All methods use ResNet-50 as backbone. **C**: additionally using COCO train2017 images that contain YouTube-VIS categories for training. The inference speed is tested on a Nvidia RTX2080Ti GPU. * indicates using deformable convolution [52] in backbone. ‡ indicates that the FPS is measured by parallel processing of images in one clip rather than sequential processing.

| Method | Online | YouTube-VIS 2019 Val. | | | | | YouTube-VIS 2021 Val. | | | | | FPS |
| | | AP | $AP_{50}$ | $AP_{75}$ | $AR_1$ | $AR_{10}$ | AP | $AP_{50}$ | $AP_{75}$ | $AR_1$ | $AR_{10}$ | |
| STEm-Seg [7] (**C**) | ✗ | 30.6 | 50.7 | 33.5 | 31.6 | 37.1 | - | - | - | - | - | 4.4 |
| VisTR [8] | ✗ | 35.6 | 56.8 | 37.0 | 35.2 | 40.2 | - | - | - | - | - | 30.0‡ |
| Propose-Reduce [9] (**C**) | ✗ | 40.4 | 63.0 | 43.8 | 41.1 | 49.7 | - | - | - | - | - | < 20 |
| MaskProp* [23] | ✗ | 40.0 | - | 42.9 | - | - | - | - | - | - | - | < 10 |
| IFC [10] | ✗ | 39.0 | 60.4 | 42.7 | 41.7 | **51.6** | 35.2 | 57.2 | 37.5 | - | - | 46.5‡ |
| EfficientVIS [37] | ✗ | 37.9 | 59.7 | 43.0 | 40.3 | 46.6 | 34.0 | 57.5 | 37.3 | **33.8** | **42.5** | 36‡ |
| MaskTrack R-CNN [1] | ✓ | 30.3 | 51.1 | 32.6 | 31.0 | 35.5 | 28.6 | 48.9 | 29.6 | 26.5 | 33.8 | 26.1 |
| SipMask [4] | ✓ | 33.7 | 54.1 | 35.8 | 35.4 | 40.1 | 31.7 | 52.5 | 34.0 | 30.8 | 37.8 | 30 |
| STMask* [20] | ✓ | 33.5 | 52.1 | 36.9 | 31.1 | 39.2 | - | - | - | - | - | 28.6 |
| SG-Net [2] | ✓ | 34.8 | 56.1 | 36.8 | 35.8 | 40.8 | - | - | - | - | - | 23.0 |
| PCAN [22] | ✓ | 36.1 | 54.9 | 39.4 | 36.3 | 41.6 | - | - | - | - | - | - |
| CrossVIS [18] | ✓ | 36.3 | 56.8 | 38.9 | 35.6 | 40.7 | 34.2 | 54.4 | 37.9 | 30.4 | 38.2 | 25.6 |
| HybridVIS [21] (**C**) | ✓ | 41.3 | 61.5 | 43.5 | **42.7** | 47.8 | 35.8 | 56.3 | 39.1 | 33.6 | 40.3 | < 20 |
| **InsPro-lite** | ✓ | 38.7 | 60.9 | 41.7 | 36.9 | 43.6 | - | - | - | - | - | 45.7 |
| **InsPro** | ✓ | 40.2 | 62.9 | 43.1 | 37.6 | 44.5 | 36.1 | 57.6 | 39.6 | 30.9 | 40.4 | 26.3 |
| **InsPro** (**C**) | ✓ | **43.2** | **65.3** | **48.0** | 38.8 | 49.0 | **37.6** | **58.7** | **40.9** | 32.7 | 41.4 | 26.3 |

## 4.2 Implementation Details

We implement our InsPro with Detectron2 [49], and most hyperparameters are set following Sparse R-CNN [12] and CondInst [39] unless otherwise specified. More implementation details can be found in Appendix A.1.

**Training Details**   We employ AdamW [50] with an initial learning rate of $2.5 \times 10^{-5}$ and weight decay 0.0001 as our model optimizer. We initialize our model with parameters pre-trained on COCO [51], and train it for 32k iterations where the learning rate is divided by 10 at iterations 24k and 28k, respectively. The training is performed end-to-end on 8 Nvidia RTX2080Ti GPUs and each GPU holds one mini-batch which contains three frame images randomly sampled from the same video. Data augmentation includes only random horizontal flip and multi-scale training where the training image is resized so that the length of its shortest side is at least 288 and at most 512. Unless otherwise noted, our InsPro adopts ResNet-50 [13] as backbone and uses 100 instance queries in our experiments.

**Inference Details**   In inference, we resize the frame image size to $640 \times 360$, which follows MaskTrack R-CNN [1]. The size of the feature bank is set to 18 by default. If the generated proposal box exceeds the frame's boundaries, it will be clipped to corresponding boundaries. No multi-scale testing is adopted in our experiments.

**InsPro-lite**   We also build a lite version of our method, named InsPro-lite. In this variant, inspired by [32], we divide video frames into key frames and non-key frames, *i.e.*, we select one key frame per $K$ frames in a video and treat other frames as non-key ones. $K$ is 10 by default. On key frames, we conduct the dynamic instance interaction 6 times while only once on non-key frames. This takes advantage of the redundancy of videos and helps reduce inference computation time. Our InsPro-lite reaches a high inference speed of 45.7 FPS at a small accuracy loss (Table 1).

## 4.3 Main Results

We perform a thorough comparison of our InsPro to state-of-the-art VIS methods on YouTube-VIS 2019 and 2021. Existing VIS methods can be divided into two categories according to whether they run online or offline [53]. Since some methods [7, 9] use 80k transformed COCO images [51] as extra training data to prevent overfitting to YouTube-VIS, for a fair comparison, we also report our

Table 2: Ablation studies on YouTube-VIS 2019.

(a) Effectiveness of instance query and proposal propagation, and temporally consistent matching (TCM).

|     | query | proposal | TCM | AP | $AP_{50}$ | $AP_{75}$ |
|-----|-------|----------|-----|------|------|------|
| (A) |       |          |     | 24.0 | 41.3 | 24.2 |
| (B) | ✓     | ✓        |     | 36.3 | 56.3 | 38.9 |
| (C) | ✓     | ✓        | ✓   | **37.4** | **57.6** | **41.1** |
| (D) | ✓     |          | ✓   | 36.7 | 57.3 | 39.9 |
| (E) |       | ✓        | ✓   | 36.6 | 55.5 | 40.3 |

(b) Effectiveness of the proposed box deduplication loss (BDL).

|        | AP | $AP_{50}$ | $AP_{75}$ | FPS |
|--------|------|------|------|------|
| w/o BDL | 37.4 | 57.6 | 41.1 | 26.3 |
| w/ BDL | **38.4** | **57.7** | **41.6** | 26.3 |

(c) Intra-query attention. $T$ is the length of the feature bank.

|       | AP | $AP_{50}$ | $AP_{75}$ | FPS |
|-------|------|------|------|------|
| T=1   | 38.4 | 57.7 | 41.6 | 26.3 |
| T=9   | 39.7 | 61.6 | 42.1 | 26.3 |
| T=18  | **40.2** | **62.9** | **43.1** | 26.3 |
| T=27  | 40.1 | 62.6 | 42.2 | 26.3 |
| T=36  | 40.1 | 62.5 | 42.2 | 26.3 |

(d) Comparison between our temporal propagation paradigm and 'tracking-by-detection' paradigm.

|                      | AP | $AP_{50}$ | $AP_{75}$ | Param (M) | FLOPs (G) | FPS |
|----------------------|------|------|------|------|------|------|
| Tracking-by-detection | 31.5 | 49.3 | 34.1 | 119.9 | 48.3 | 25.4 |
| Ours                 | **37.4** | **57.6** | **41.1** | **106.1** | **45.5** | **26.3** |

(e) Comparison between our united query and 'track-and-detect query'.

|                      | AP | $AP_{50}$ | $AP_{75}$ |
|----------------------|------|------|------|
| Track-and-detect query | 37.4 | 56.9 | 40.3 |
| Ours                 | **38.4** | **57.7** | **41.6** |

results with and without extra COCO training data. Table 1 presents all the results obtained with a ResNet-50 backbone on a Nvidia RTX2080Ti GPU.

**YouTube-VIS 2019**   Table 1 (**left**) shows the comparison between our InsPro and other state-of-the-art methods on YouTube-VIS 2019 validation set. We can see that, in the online group, our InsPro outperforms all other popular methods under the same data setting. Specifically, our InsPro achieves 40.2 AP without COCO data and 43.2 AP with COCO data respectively, surpassing other online VIS methods by a large margin. Even our lite version, InsPro-lite, performs better than all other online methods trained without COCO data, reaching 38.7 AP at an impressive speed of 45.7 FPS.

**YouTube-VIS 2021**   Table 1 (**right**) displays results on YouTube-VIS 2021 validation set. It shows a similar comparison pattern to YouTube-VIS 2019 and our InsPro achieves the state-of-the-art performance once again.

## 4.4   Ablation Study

We conduct extensive experiments on YouTube-VIS 2019 to study the effectiveness and individual performance contribution of our proposed modules.

**Temporal Propagation and Matching**   Our InsPro is built on the proposed instance query and proposal temporal propagation mechanism. Table 2 (a) shows how this mechanism contributes to our high performance. In this table, method A represents the video instance segmentation baseline, where each frame is processed individually without any temporal propagation, and object instances generated on each frame are linked if they are produced from the same instance query slice. Since this method lacks the mechanism to ensure the instance query-proposal pair corresponds to the same object across frames, it only achieves 24.0 AP due to inaccurate instance association. By contrast, when we add the temporal propagation (method B), we can easily improve the performance significantly to 36.3 AP. This evidences the importance and effectiveness of the proposed temporal propagation technique in a query-based VIS framework. If we further adopt the temporally consistent matching strategy during training (method C), we achieve an even better performance of 37.4 AP.

We also analyze the separate performance of propagating only instance query (method D) or instance proposal (method E). The results show that these two settings achieve a similar performance boost (36.7 AP vs 36.6 AP). Applying them together yields 37.4 AP, bringing further performance gain.

**Box Deduplication Loss**   We propose a box deduplication loss to suppress the duplicate proposal boxes on the same object across frames. With the qualitative results shown in Figure 3, we show the quantitative comparison in Table 2 (b). We can see that supervising the learning with this loss during training can lead 1.0 AP improvement (38.4 AP vs 37.4 AP). This performance gain is brought by fewer duplicate boxes and fewer missed detections.

**Intra-query Attention**  We perform intra-query attention inside a feature bank to augment the instance query so that it can capture long-range temporal cues. As we can see in Table 2 (c), this simple method works well and improves the performance considerably. In particular, $T = 1$ indicates no intra-query attention is used and 38.4 AP is achieved. When we increase the volume $T$ of the feature bank, the performance rises and saturates at 40.2 AP with $T = 18$. It is worthwhile to note that this lightweight yet effective intra-query attention module brings almost no speed drop.

**Temporal Propagation *vs*. Tracking-by-Detection**  Despite the fact that our InsPro does not perform explicit instance association, it still outperforms all other online methods implementing explicit tracking or matching. To verify that our superior performance comes from the temporal propagation mechanism rather than the instance segmentation model design, we compare our temporal propagation VIS approach to the typical 'tracking-by-detection' paradigm with the same instance segmentation baseline. We implement a 'tracking-by-detection' VIS system by replacing the Mask R-CNN part in MaskTrack R-CNN [1] with our instance segmentation model. In this case, the only independent variable is the object tracking method.

As shown in Table 2 (d), our InsPro surpasses the 'tracking-by-detection' model by a large margin even if our design is simpler and faster, which soundly proves the effectiveness of our method. We argue again that this is because the evolving instance query-proposal pair in propagation encodes object temporal and spatial cues intrinsic in videos, whereas 'tracking-by-detection' methods are generally incapable of exploiting this advantage.

**Our United Query *vs*. Track-and-Detect Query**  We further compare our method to those MOT methods that adopt a similar query-propagation method for object tracking. These methods rely on two different query sets, *i.e.*, a track query set and an object query set, to track seen objects and detect new objects respectively, while we only maintain one united query set. They need heuristic rules to combine these two type queries. Meanwhile, they manually select track queries with high scores from the previous frame to build the track query set. This makes them complex and less effective in tracking since occluded objects with low scores probably have broken trajectories because of the filtering.

To show the superiority of our method, we compare the 'track-and-detect query' paradigm adopted in the most recent MOTR [16] to ours using the same instance segmentation baseline. We follow MOTR [16] exactly to set up the model and experiment settings. To exclude the influence of other factors, we do not use temporal feature aggregation in both methods. Table 2 (e) shows the comparisons on YouTube-VIS 2019. It can be seen that our InsPro achieves a higher performance even using a simpler query design. We attribute this advantage to our conscientiously designed modules described in Sec 3.

## 5  Conclusion

In this paper, we propose a simple, fast and yet effective query-based framework for online VIS. In this framework, we rely on a novel instance query and proposal propagation mechanism to undertake VIS, where we generate object instances based on a set of evolving instance query-proposal pairs propagated from previous frames. This mechanism enables our model not only to associate object instances implicitly, but to utilize video temporal cues elegantly. To make this propagation mechanism work well, we develop several modules to ensure that the learned instance query-proposal pair keeps being bound to one object, These modules include an intra-query attention unit, a temporally consistent matching mechanism and a box deduplication loss. Extensive experiments on YouTube-VIS 2019 and 2021 verify the effectiveness of our designs, and show that our InsPro achieves superior VIS performance, outperforming all other online VIS methods.

## Acknowledgments

This work is supported in part by the National Natural Science Foundation of China (Grant No. 61721004), the Projects of Chinese Academy of Science (Grant No. QYZDB-SSW-JSC006), the Strategic Priority Research Program of Chinese Academy of Sciences (Grant No. XDA27000000), and the Youth Innovation Promotion Association CAS.

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
