# A Appendix

## A.1 More Details about InsPro-lite

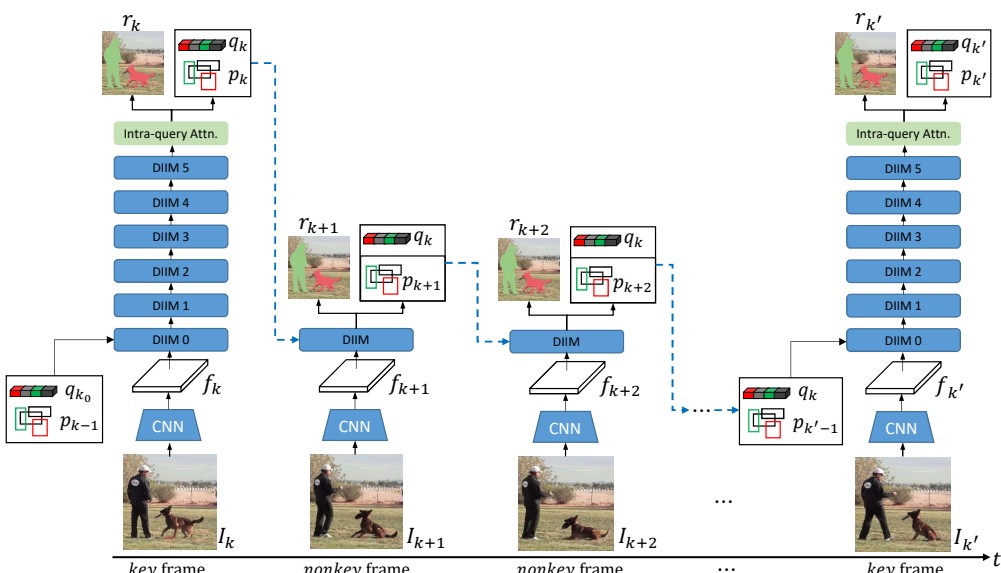

Figure 4: Overview of our InsPro-lite.

**Overview**   Figure 4 displays the overview of our InsPro-lite. In this framework, the video frames are divided into key frames and non-key frames, and we process them differently. On the key frame $I_k$, the segmentation head consists of 6 dynamic instance interaction modules (DIIM) and one intra-query attention module for generating VIS results $r_k$, and both the instance query and proposal are updated. On the non-key frame $I_{k+1}$, the segmentation head contains only one dynamic instance interaction module to generate VIS results $r_{k+1}$, and only the instance proposal is updated while the propagated instance query keeps unchanged. For more details, please refer to [32].

**Training Details**   Two-phase training is performed for InsPro-lite. In the first phase, we train the segmentation head processing key frames as described in Sec. 4.2. In the second phase, We train the segmentation head processing non-key frames while other parts in the network are fixed. Each training batch contains five frame images randomly sampled from the same video, with three as key frames and two as non-key frames.

**Effects of key frame interval**   Table 3 lists the performances of InsPro-lite using different key frame intervals. When $K = 1$, it represents the original InsPro model.

Table 3: Performances of InsPro-lite (ResNet-50 backbone) using different key frame intervals. The inference speed is tested on a Nvidia RTX2080Ti GPU.

| $K$ | 1 | 5 | 10 | 15 |
|-----|------|------|------|------|
| AP | 40.2 | 39.4 | 38.7 | 37.5 |
| FPS | 26.3 | 41.8 | 45.7 | 49.1 |

## A.2 Additional Comparison on OVIS

We additionally evaluate our InsPro on the occluded video instance segmentation dataset OVIS [54]. OVIS is a very challenging dataset that contains many occlusion scenes. It consists of 296k high-quality instance masks (about $2\times$ of YouTube-VIS 2019 [1]), 5.8 instances per video (about $3.4\times$ of YouTube-VIS 2019), 25 semantic categories, 607 training videos and 140 validation videos.

Table 4 shows the comparison between our InsPro and other state-of-the-art methods on the OVIS validation set. Due to the severe occlusion and crowded scenes, the performances of all methods

Table 4: Comparison of our InsPro to state-of-the-art methods on the OVIS validation. All methods use ResNet-50 [13] as backbone. The inference speed is tested on a Nvidia RTX2080Ti GPU.

| Method | Online | AP | $AP_{50}$ | $AP_{75}$ | $AR_1$ | $AR_{10}$ | FPS |
|---|---|---|---|---|---|---|---|
| STEm-Seg [7] | ✗ | 13.8 | 32.1 | 11.9 | 9.1 | 20.0 | 4.4 |
| MaskTrack R-CNN [1] | ✓ | 10.9 | 26.0 | 8.1 | 8.3 | 15.2 | 26.1 |
| MaskTrack R-CNN [1] + Calibration [54] | ✓ | 14.9 | 32.4 | 12.5 | 9.1 | 19.5 | < 26.1 |
| SipMask [4] | ✓ | 10.3 | 25.4 | 7.8 | 7.9 | 15.8 | 30 |
| SipMask [4] + Calibration [54] | ✓ | 13.9 | 30.7 | 11.9 | 9.4 | 19.4 | < 30 |
| CrossVIS [18] | ✓ | 14.9 | 32.7 | 12.1 | - | - | 25.6 |
| CrossVIS [18] + Calibration [54] | ✓ | 18.1 | 35.5 | 16.9 | - | - | < 25.6 |
| **InsPro** | ✓ | **21.1** | **42.6** | **19.0** | **11.1** | **25.6** | 26.3 |

Table 5: Comparison between InsPro and other object instance segmentation baselines on COCO validation set. All methods adopt ResNet-50 backbone, a training time of 36 epochs, and the same data augmentation. QueryInst [3] and InsPro both use 100 queries.

| Methods | AP | $AP_{50}$ | $AP_{75}$ | $AP_S$ | $AP_M$ | $AP_L$ |
|---|---|---|---|---|---|---|
| Mask R-CNN [5] | 37.5 | 59.3 | 40.2 | 21.1 | 39.6 | 48.3 |
| QueryInst [3] | 39.8 | 61.8 | 43.1 | 21.3 | 42.7 | 58.3 |
| InsPro | 39.4 | 61.8 | 41.9 | 19.7 | 42.9 | 59.3 |

on OVIS drop significantly compared to those on YouTube VIS [1]. To tackle occlusion, previous methods [1, 4, 18] apply temporal feature calibration [54] to align adjacent frames and complement the mission objective cues in severe occlusion, which increases the overhead of these systems. Our InsPro employs flexible instance query-proposal pair propagation and intra-query attention, outperforming all other popular methods without feature calibration.

### A.3 Object Instance Segmentation Baselines Comparison

The performance of the object instance segmentation baseline is also one of the important factors affecting the VIS results. Table 5 lists the performance comparison between our InsPro, Mask R-CNN [5] and QueryInst [3] on the COCO instance segmentation validation set. Except that they use different segmentation head structures, both of them adopt the same ResNet-50 backbone, same training time of 36 epochs, and the same data augmentation with ours. QueryInst [3] and InsPro both use 100 queries. It can be seen that our base model performs a bit poorer than QueryInst.

### A.4 Qualitative Results

We show qualitative results under some challenging scenarios on YouTube-VIS 2019 [1] and OVIS [54] in Figure 5 and Figure 6, respectively.

Our InsPro is robust to many challenging situations, including fast motion, occlusion, crowd, similar objects, and motion blur, *etc*. For example, our InsPro can easy handle new objects with similar appearance, as shown in line 6 and 8 of Figure 5. This is because our InsPro propagates not only object queries but their corresponding proposals. Since those proposals have tracked objects positional prior encoded, when using such a query-proposal pair to predict objects, it is easy for the network to distinguish objects of similar appearance. However, InsPro may fail to segment some small objects since it has no specific design for processing small objects.

### A.5 Additional Experiments on ImageNet VID

To further verify the effectiveness and generality of our InsPro, we conduct additional experiments on ImageNet VID [55]. ImageNet VID [55] is a large-scale video dataset where the object instances on every frame are fully annotated with object bounding box, object category and instance ID. It consists of 3,862 training videos and 555 validation videos from 30 object categories. The average length of videos in ImageNet VID is 317 (about $\times$ 11.5 of YouTube-VIS 2019 [1]). We evaluate the joint detection and tracking performance of our InsPro on ImageNet VID. Following [1], we use video-level average precision (AP) as the evaluation metric and the video-level IoU is computed using box sequences instead of mask sequences.

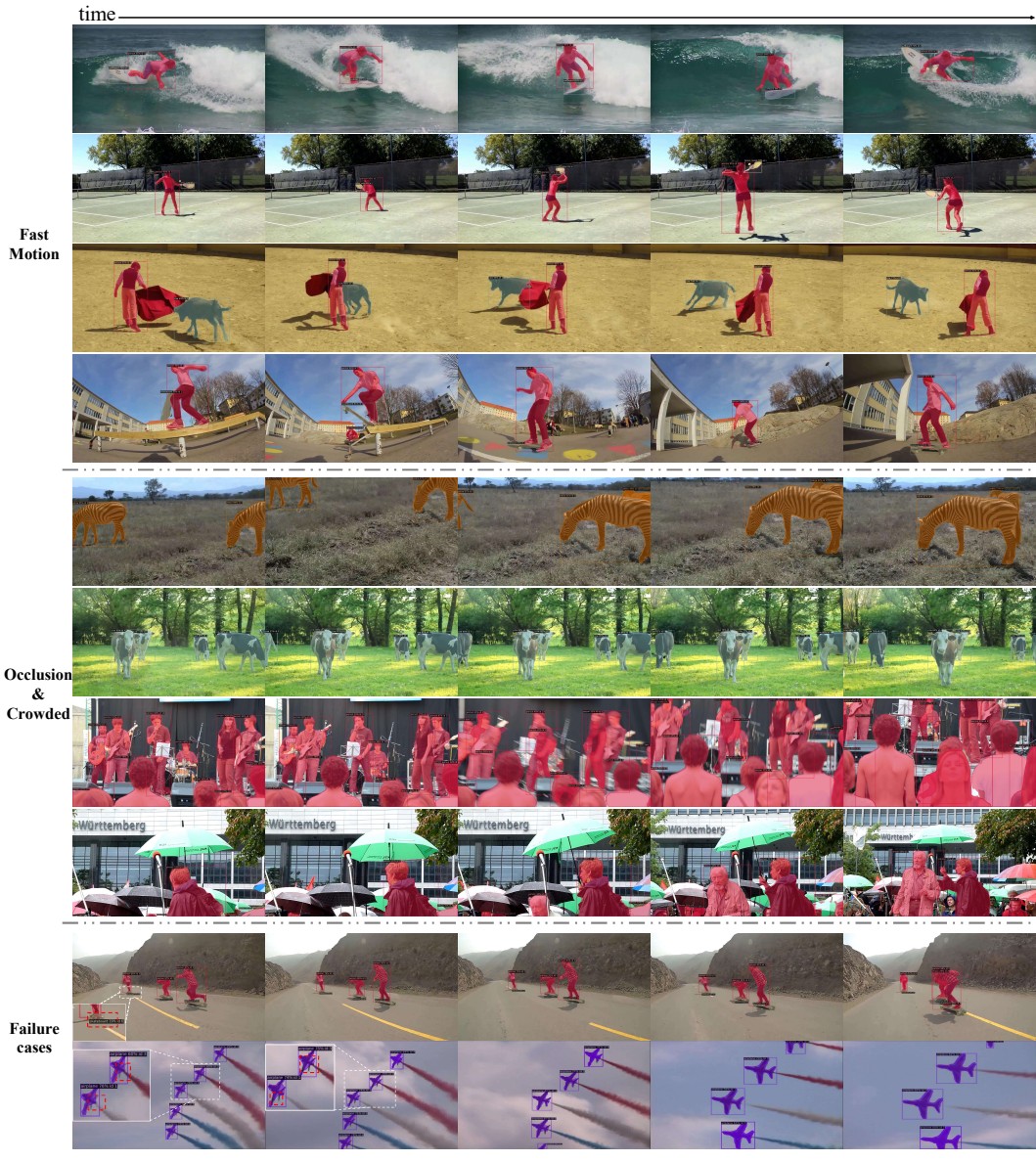

Figure 5: Qualitative results on the YouTube-VIS 2019 [1] validation set. Our InsPro with ResNet-50 [13] backbone performs well under many challenging scenes, including fast motion, occlusion and crowded. InsPro may fail to segment some small objects since it has no specific design for processing small objects.

### A.5.1 Implementation Details

We make minor modifications to InsPro to adapt it for joint detection and tracking on ImageNet VID. We remove the segmentation part in the instance segmentation module (Sec. 3.2), and the rest remain unchanged. During training, the segmentation losses $\mathcal{L}_{dice}$ and $\mathcal{L}_{focal}$ are removed from the overall loss $\mathcal{L}$ (Equation 6) either. The modified InsPro, called InsPro-VID, is trained on both the ImageNet DET training set and the ImageNet VID training set. Each training batch contains three frame images. On VID, the three images are randomly sampled from the same video, while on DET, the three images are the same because DET contains only images. InsPro-VID is trained for 90k iterations and the initial learning rate is set to $2.5 \times 10^{-5}$, which is divided by 10 at iteration 65k and 80k, respectively.

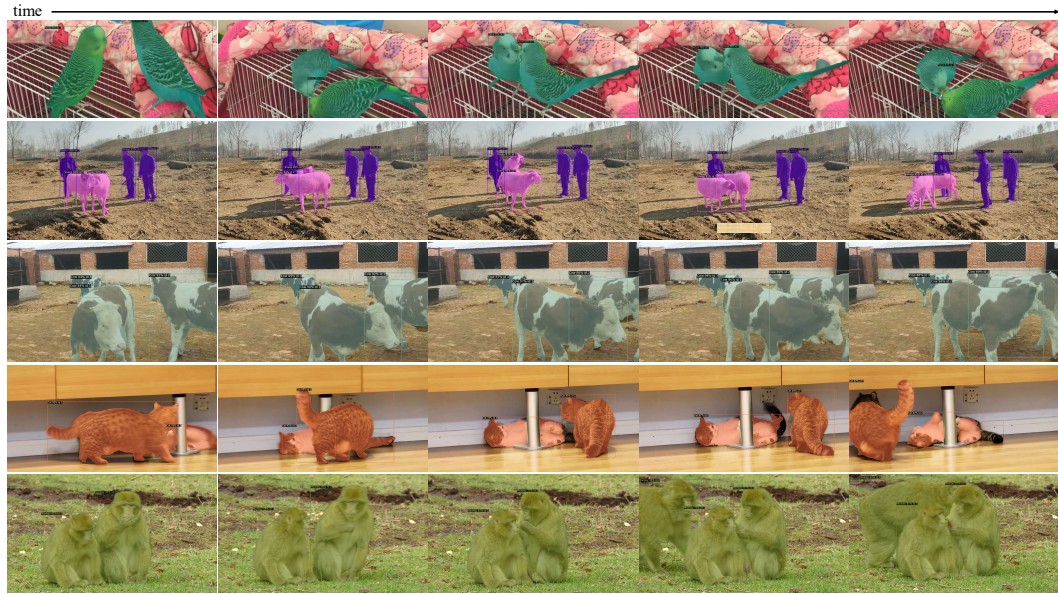

Figure 6: Qualitative results on the challenging OVIS [54] validation set.

Table 6: Main results and ablation studies on ImageNet VID [55].

(a) Comparison to other methods. All methods use ResNet-101 [13] as backbone.

| Methods | $AP_{50}$ |
|---|---|
| Faster R-CNN [31] + Viterbi [56] | 60.3 |
| D&T [57] | 60.5 |
| STMN [58] + Viterbi [56] | 60.4 |
| **InsPro-VID** | **64.1** |

(b) Effectiveness of the temporal propagation, and temporally consistent matching (TCM).

|  | propagation | TCM | AP | $AP_{50}$ | $AP_{75}$ |
|---|---|---|---|---|---|
| (A) |  |  | 9.7 | 15.7 | 9.1 |
| (B) | ✓ |  | 29.2 | 43.8 | 30.6 |
| (C) | ✓ | ✓ | **33.2** | **48.8** | **35.5** |

(c) Effectiveness of the proposed box deduplication loss. $\beta$ is a hyperparameter in Equation 4.

|  | AP | $AP_{50}$ | $AP_{75}$ |
|---|---|---|---|
| $\beta$=0 | 33.2 | 48.8 | 35.5 |
| $\beta$=0.08 | 37.8 | 57.5 | 40.0 |
| $\beta$=0.1 | 38.9 | **57.8** | 42.6 |
| $\beta$=0.15 | **39.5** | 57.2 | 42.7 |
| $\beta$=0.2 | 39.2 | 57.3 | 42.7 |
| $\beta$=0.25 | 39.1 | 57.2 | **43.0** |

(d) Effectiveness of the intra-query attention. $T$ is the length of the feature bank.

|  | AP | $AP_{50}$ | $AP_{75}$ |
|---|---|---|---|
| T=20 | 39.8 | 59.1 | 43.5 |
| T=40 | 41.3 | 59.9 | 45.7 |
| T=60 | 41.8 | **60.7** | 45.9 |
| T=80 | 41.3 | 60.0 | 45.3 |
| T=100 | **42.2** | 60.6 | **46.8** |
| T=120 | 41.8 | 60.1 | 46.2 |

### A.5.2 Main Results

We compare our InsPro-VID with other methods in Table 6 (a). Other methods mainly use tracking or post-processing matching (e.g., Viterbi [56] algorithm) techniques to associate detection results across frames, which are more complex and unable to fully utilize temporal cues. Our InsPro-VID achieves instance association in parallel with detection and elegantly takes advantage of temporal clues in videos through the proposed instance query-proposal propagation mechanism. As shown in Table 6 (a), our InsPro-VID achieves superior performance compared over other methods.

### A.5.3 Ablation Study

We conduct extensive experiments on ImageNet VID to study the effectiveness and individual performance contribution of our proposed modules. All experiments use ResNet-50 [13] as backbone.

**Temporal Propagation and Matching**  Table 6 (b) shows how the proposed instance query and proposal temporal propagation mechanism and the temporally consistent matching strategy contribute to our high performance.

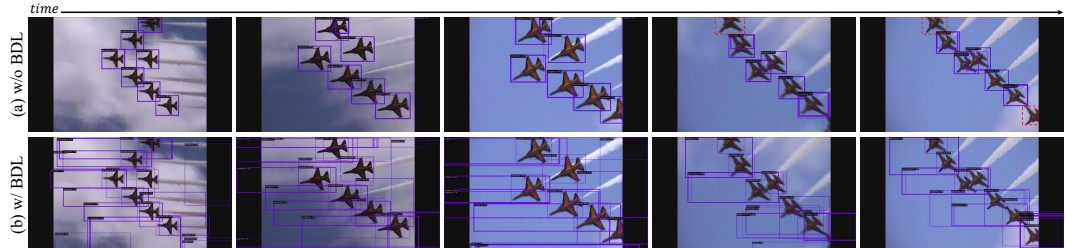

Figure 7: Visualization of box deduplication loss effects on ImageNet VID. We display all predicted boxes on each frame without a score threshold, with 100 detections by default. (a) All predicted boxes in the first three frames are clustered around a few instances if the box deduplication loss is not applied, and newly emerging instances (shown in red box) on the last two frames cannot be detected. (b) After applying the box deduplication loss (BDL), each instance is predicted by only one accurate predicted box and the missing new objects in the first row are detected.

**Box Deduplication Loss**   Table 6 (c) shows the effectiveness of the proposed box deduplication loss. $\beta$ is a hyperparameter in Equation 4, which controls the center distance between the duplicate predicted boxes and the ground truth box during training. $\beta = 0$ indicates no box deduplication loss is used during training and 33.2 AP is achieved. When we increase the value of $\beta$, the performance rises and saturates at 39.5 AP when $\beta = 0.15$.

We also provide some qualitative results in Figure 7 to show the effect of the box deduplication loss. We display all predicted boxes on each frame without a score threshold, with 100 detections by default. The first row is the visualization results when $\beta = 0$. We can see that all predicted boxes in the first three frames are clustered around a few instances. Since in our temporal propagation mechanism the prediction of the next frame is generated based on the instance proposals propagated from the last frame, if all predicted boxes in the last frame are clustered around the existing instances, it will be difficult for the model to detect new objects in the next frame. As shown in Figure 7 (a), we can see that the newly emerging instances in red box on the last two frames are indeed not detected. Figure 7 (b) is the visualization results when $\beta = 0.15$. It shows that each instance has only one accurate predicted box and the new emerging objects are detected.

Therefore, the proposed box deduplication loss enables our InsPro not only to alleviate the duplicate problem, but to detect new objects easily with the sparsely distributed unmatched proposal.

**Intra-query Attention**   Table 6 (d) shows the effectiveness of the intra-query attention. When increasing the size $T$ of the feature bank, the instance query can capture more temporal cues and the performance improves considerably.

### A.6   Broader Impact and Future Work

Our InsPro introduces a novel instance query and proposal propagation mechanism to the VIS system, and constructs a simple, fast and yet effective online framework to achieve one-shot video instance segmentation. Our InsPro-lite can achieve promising accuracy while running in real-time. Due to its fine-grained object representation result and efficiency, we believe our VIS system InsPro can positively impact many applications such as autonomous driving and video editing, *etc*.

For future work, we plan to verify the generality of our method in other query-based frameworks [11, 25, 59], since our proposed temporal propagation mechanism is only applied in Sparse R-CNN [12] at present. Furthermore, we also intend to implement the simple yet effective temporal propagation mechanism in tasks that require instance association, such as multi-object tracking [60], panoptic segmentation [61, 62] in video, and semi-supervised video object segmentation [63].