# OpenReview forum: "InsPro: Propagating Instance Query and Proposal for Online Video Instance Segmentation"
_NeurIPS.cc/2022/Conference — NeurIPS 2022 Accept_

### Official Review · Reviewer_1ynk · 2022-07-09

**Rating:** 6
**Confidence:** 5
**Soundness:** 3 good
**Presentation:** 3 good
**Contribution:** 3 good

**Summary:**

This paper proposes a query-based framework for online video instance segmentation, which designs an instance query and proposal propagation mechanism to perform instance association implicitly. With such a mechanism, they achieve implicit instance association in parallel with segmentation and elegantly take advantage of temporal clues in videos. Experiments show the effectiveness of this method.

**Questions:**

As described in weakness.

**Limitations:**

No serious negative societal impact of this work.

**Strengths And Weaknesses:**

Strengths:
1) This paper proposes a novel query&proposal propagation mechanism for the video instance segmentation task, which is proved effective.
2) This paper achieves satisfactory results among the online VIS methods.
3) This paper is well organized.

Weakness:
1) The key idea of query&proposal propagation in this paper is similar to the method proposed in TrackFormer [28] (CVPR 2022), thus the author may claim the relation and difference between them.
2) It is confusing for the baseline of ablation study in Tab. 2 (a)(c)(d). The AP score of 37.4 in these tables is mismatched with the AP score of 40.2 in Tab. 1 and Tab. 2(b), thus the author may clarify the setting or baseline for these results.
3) It would be better to have an analysis of the variance of InsPro-lite, e.g. the effects of key frame numbers.

---

> ### Author Response · Authors · 2022-08-02
> **Rebuttal**
>
> We thank the reviewer for appreciating our work and giving helpful advice. In what follows, we deal with the concerns.
> #### __Weaknesses__:
> __Q1__: The author may claim the relation and difference between InsPro and TrackFormer. \
> __A1__: Please refer to the __Common Concern__ (https://openreview.net/forum?id=V3kqJWsKRu4&noteId=B1zWW1R5IvR).
>
> __Q2__: The author may clarify the setting or baseline for these results in ablation studies. \
> __A2__: The table below lists the experiment settings used for the ablation studies in Table 2. __Table 2(a)__ shows how the proposed instance query-proposal propagation and temporally consistent matching contribute to the performance. __Table 2(a) C__ represents the basic version of InsPro, which achieves an AP of 37.4. __Table 2(c)__ displays the effectiveness of the proposed box deduplication loss (BDL). Equipping the basic InsPro with BDL, the AP score is improved from 37.4 to 38.4. __Table 2(b)__ shows the effectiveness of the intra-query attention. After adding the intra-query attention to the w/BDL method in Table 2(c), the AP score is further improved from 38.4 to 40.2, and we use 40.2 AP as the final InsPro performance for comparison in Table 1(a).
> Besides, __Table 2(d)__ shows the superiority of our proposed temporal propagation strategy over the commonly used 'track-by-detect' paradigm (37.4 AP vs 31.5 AP). To ensure a fair comparison, we restrict the only independent variable in this experiment to be the object tracking method and more details can be found in Sec. 4.4.
> Thank you for pointing out this confusion. We will correct it in our new version.
>
> Methods             |                 experimental setting                 |    AP    |   AP50   |   AP75   |
> ---                 |:----------------------------------------------------:|:--------:|:--------:|:--------:|
> __Tab.2(a) A__      |         image instance segmentation baseline         |   24.0   |   41.3   |   24.2   |
> __Tab.2(a) B__      | __Tab.2(a) A__ + instance query-proposal propagation |   36.3   |   56.3   |   38.9   |
> __Tab.2(a) C__      |   __Tab.2(a) B__ + temporally consistent matching    | __37.4__ |   57.6   |   41.1   |
> __Tab.2(c) w/ BDL__ |    __Tab.2(a) C__ + box deduplication loss (BDL)     |   38.4   |   57.7   |   41.6   |
> __Tab.2(b) T=18__   |     __Tab.2(c) w/ BDL__ + intra-query attention      | __40.2__ | __62.9__ | __43.1__ |
> __Tab.2(d) Track-by-detect__ |          __Tab.2(a) A__ + explicit tracking          |   31.5   |   49.3   |   34.1   |
>
> __Q3__: It would be better to have an analysis of the variance of InsPro-lite, e.g. the effects of key frame numbers. \
> __A3__: Thank you for this suggestion. The table below lists the performances of InsPro-lite using different key frame intervals (GPU: 1 RTX 2080Ti). When k=1, it represents the original InsPro model. In addition, we provide more details about InsPro-lite in Sec. A.1 of the supplementary material.
>
> k   |  1   |  5   |  10  |  15  |
> --- |:----:|:----:|:----:|:----:|
> AP  | 40.2 | 39.4 | 38.7 | 37.5 |
> FPS | 26.3 | 41.8 | 45.7 | 49.1 |

---

> > ### Author Response · Authors · 2022-08-09
> > **Thanks for your review.**
> >
> > Thanks for your review, which makes our paper more clear and sound. We hope that we have addressed all your concerns.
> >
> > As it is the last day of the Reviewer-Author Discussion session, if you have other concerns, please don't hesitate to let us know.

---

### Official Review · Reviewer_o3dJ · 2022-07-10

**Rating:** 5
**Confidence:** 5
**Soundness:** 2 fair
**Presentation:** 3 good
**Contribution:** 2 fair

**Summary:**

The paper proposes InsPro for online video instance segmentation. Instance queries are propagated and updated from the previous frames to current frame with implicit object association. Intra-query attention, temporally consistent matching, and box deduplication loss
are also proposed. Experiments are conducted on the Youtube-VIS 2019 and 2021.


**Questions:**

I will consider raise my rating if the concern in the weakness part can be well addressed. The paper neglects discussion/comparison with related works in query-based VIS and temporal object propagation. The key idea of query propagation has been explored in TransTrack, which influences the paper novelty a lot.

**Limitations:**

There is no limitation or potential negative societal impact discussion in the paper.

**Strengths And Weaknesses:**

**Strengths**:

1. The paper has a good motivation to solve online VIS problem with implicit query matching. The proposed method achieves both good evaluation performance and inference speed.

2. The paper shows the effect of the proposed query/proposal propagation, intra query attention and BDL with adequate ablation experiments.

3. The paper is organized well with good writing and figures.

**Weakness**:

1. Missing discussion for related works in query-based VIS methods. In the paragraph for Query-based Methods, both IFC[a] and VisTr [b] are neglected. Also, SeqFormer[c] and Mask2Former[d] are also query-based VIS although they are arXiv but worth mentioning.

2. Missing related works in clip-level VIS Methods, where the online VIS method PCAN [e] also proposes the online object feature propagation and updating for the target tracklet. The tech differences comparison to [e] should be discussed in the related work section as well as the table results comparison on Youtube-VIS.

[a] Video instance segmentation using inter-frame communication transformers. NeurIPS, 2021.

[b] End-to-end video instance segmentation with transformers. CVPR, 2021.

[c] SeqFormer: a Frustratingly Simple Model for Video Instance Segmentation. arXiv:2112.08275

[d] Mask2former for video instance segmentation. arXiv preprint arXiv:2112.10764 (2021)

[e] Prototypical Cross-Attention Networks for Multiple Object Tracking and Segmentation. NeurIPS, 2021.


3. The tech contribution of InsPro is limited. The idea of query propagation has been adopted in both TransTrack and MOTR. The frame-level InsPro is a straightforward combination between Sparse R-CNN (for query-based object detection) and CondInst(providing mask head). Then, the frame-level InsPro is further extended to video by instance query propagation during online inference.

4. What are the typical failure cases of the methods? How to handle new objects with similar appearance? How to handle/correct the accumulation error during propagation process, for example, an instance query is wrongly matched at an early inference stage?

---

> ### Author Response · Authors · 2022-08-02
> **Rebuttal**
>
> Thanks the reviewer for approving of the motivation and performance of our work, and giving useful advice. We address the concerns as follows.
> #### __Weaknesses__:
> __Q1__: Missing discussion of related query-based VIS methods and clip-level VIS methods. \
> __A1__: We appreciate the reviewer's extensive knowledge of VIS. We will add more discussion of IFC and VisTR in the Query-based methods subsection, and include SeqFormer and Mask2Former in this part too.
>
> __Q2__: The tech differences comparison to PCAN. \
> __A2__: Thanks for this suggestion. We will add the following discussion of PCAN and performance comparison to our revised version.
> PCAN proposes frame- and instance-level prototypical cross-attention modules to leverage rich spatio-temporal information distilled from previous frames to facilitate better segmentation. This generates augmented features for model to produce better object instances in the current frame, which shares the similar benefits of our intra-query attention module.  However, PCAN adopts an additional explicit tracking head to complete object association, while our InsPro performs the association implicitly through an instance query and proposal propagation mechanism, which is simpler.
> As for the performance comparison, under similar experiment settings, PCAN achieves 36.1 AP on the YouTube VIS 2019 validation set, while our InsPro reaches 40.2 AP.
>
> __Q3__: The novelty of InsPro and the discussion among InsPro, TransTrack, and MOTR. \
> __A3__: Please refer to the __Common Concern__ (https://openreview.net/forum?id=V3kqJWsKRu4&noteId=B1zWW1R5IvR).
>
> __Q4__: What are the typical failure cases of the methods? \
> __A4__: From our observation, InsPro may fail to segment and track some tiny objects since we do not make special designs for handling tiny objects.
>
> __Q5__: How to handle new objects with similar appearance? \
> __A5__: This is a good question. In our system, we propagate not only object queries but their corresponding proposals. Since those proposals have tracked objects positional prior encoded, when using such a query-proposal pair to predict object, it is easy for the network to distinguish objects of similar appearance. Please refer to Figure 3 and Figure 5 in the supplementary material for some examples.
>
> __Q6__: How to handle/correct the accumulation error during propagation process . For example, an instance query is wrongly matched at an early inference stage? \
> __A6__: We guess the reviewer is concerned that an instance query may be matched to a different object in the next frame and this error will be propagated to further frames. If so, we would like to explain that, since we do not perform explicit matching, such an error would not be propagated to further frames, and because we do instance segmentation on each frame based on the enhanced queries with their past ones by intra-query attention, this error can probably be corrected in the next frame.
>
> #### __Limitations__:
> __Q__: There is no limitation or potential negative societal impact discussion in the paper. \
> __A__: Thanks for this reminder. We have discussed the broader impact and future works in Sec. A.6 of supplementary material. We will add more limitation and societal impact discussions in the revised revision.

---

> > ### Comment · Reviewer_o3dJ · 2022-08-03
> > **Is the revised paper version available?**
> >
> > Thanks for the response. The authors promise revisions for solving the neglected discussion/comparison with related works in query-based VIS and temporal object propagation. Also, authors provides long answer in the 'common concern' with TransTrack, TrackFormer, MOTR, EfficientVIS. This is not a vey small modification but critical to the paper.
> >
> > I would appreciate a revised version to help my final rating decision. Only by clearly discussing the differences with related works can the paper novelty/advantage be revealed.

---

> > > ### Author Response · Authors · 2022-08-04
> > > **We are working on it.**
> > >
> > > Thanks for the suggestion.
> > > We have started revising the paper. Hopefully, we can complete a draft revision within this discussion period.

---

> > > > ### Comment · Reviewer_o3dJ · 2022-08-08
> > > > **Comment on the revised version**
> > > >
> > > > Thanks for the updated content. Part of my concern on the paper novelty in object query propagation, and differences to existing methods has been addressed. Thus I decide to raise my score to borderline accept.

---

> > > > > ### Author Response · Authors · 2022-08-09
> > > > > **Thanks for your review and rating upgrade.**
> > > > >
> > > > > It is glad to know that part of your concerns is addressed.
> > > > >
> > > > > Thank you for your careful review, which helps improve our paper greatly, and this kind rating upgrade. It is really encouraging and delightful! If you have any other questions, please let us know.

---

### Official Review · Reviewer_g8FE · 2022-07-11

**Rating:** 6
**Confidence:** 4
**Soundness:** 3 good
**Presentation:** 3 good
**Contribution:** 2 fair

**Summary:**

This paper presents InsPro, a new method for online video instance segmentation by propagating instance queries across frames. The proposed InsPro, built on SparseR-CNN, defines a fixed set of instance queries and proposals to recognize and segment objects, and then propagate the updated queries and proposals to next frame. A temporal memory bank along with an intra-query attention are adopted to augment the instance queries. The proposed method InsPro achieves good results on Youtube VIS-2019 and Youtube VIS-2021 in terms of online video instance segmentation.

**Questions:**

1. The proposed method is built based on Sparse R-CNN but applies dynamic/conditional convolution to generate instance masks. I'm concerned about the pretraining performance on COCO. QueryInst[1] also extends Sparse R-CNN for (video) instance segmentation, how about the performance difference between the proposed InsPro and QueryInst in terms of COCO instance segmentation?

  [1] Fang et.al. Instances as Queries.


**Strengths And Weaknesses:**

### Strength

- The proposed method based on Sparse R-CNN adopts query-proposal propagation across frames for video instance segmentation and a temporal memory bank to augment queries with past frames.
- The proposed method is simple and effective, and easy to follow.
- This paper proposes an effective box de-duplication loss to further remove the duplicates.
- The overall performance of the proposed InsPro is good and experiments are abundant.

### Weakness

- The core idea of InsPro about using query propagation across frames has been explored in several works [1][2]. TrackerFormer[1] and EfficientVIS[2] adopt a similar way to propagate queries across frames and also can be reshaped into an online method. I'm concerned about the novelty of the proposed InsPro and the detailed comparisons among InsPro, TrackerFormer, and EfficientVIS.
- Line269, Sec 4.3: As far as I know, IFC can run near-offline(T=2) inference, see [3].


[1] Meinhardt et.al. TrackFormer: Multi-Object Tracking with Transformers.
[2] Wu et.al. Efficient Video Instance Segmentation via Tracklet Query and Proposal.
[3] Hwang et.al. Video Instance Segmentation using Inter-Frame Communication Transformers.

---

> ### Author Response · Authors · 2022-08-02
> **Rebuttal**
>
> We appreciate that the reviewer thinks our method is simple, effective and easy to follow, and provides helpful comments.
>
> #### __Weaknesses__:
> __Q1__: The novelty of the proposed InsPro and the detailed comparisons among InsPro, TrackFormer, and EfficientVIS. \
> __A1__: Please refer to the __Common Concern__ (https://openreview.net/forum?id=V3kqJWsKRu4&noteId=B1zWW1R5IvR).
>
> __Q2__: IFC can run near-online inference. \
> __A2__: Thanks to the reviewer for this comment. We have noticed that a near-online inference scenario is defined in IFC to make a trade-off between latency and accuracy. In this setting, when T=5, IFC achieves 39.0 AP, which is lower than our 40.2 AP.  We will update our expression in the revised version.
>
> #### __Questions__:
> __Q__: How about the performance difference between the proposed InsPro and QueryInst in terms of COCO instance segmentation? \
> __A__: The table below lists the performance comparison between our InsPro and QueryInst on the COCO instance segmentation validation set. Except that they have different segmentation head structures, both methods adopt the same ResNet-50 backbone, 100 queries, a training time of 36 epochs, and the same data augmentation. The number of FLOPs is tested with an input image resolution of 640 x 360. It can be seen that our base model actually performs a bit poorer than QueryInst.
>
> Method                          |  AP  | AP50 | AP75 | APs  | APm  | APl  | Param (M) | FLOPs (G) |
> ---                             |:----:|:----:|:----:|:----:|:----:|:----:|:---------:|:---------:|
> QueryInst-R50-100-query-3x      | 39.8 | 61.8 | 43.1 | 21.3 | 42.7 | 58.3 |   170.8   |   95.7    |
> __InsPro-R50-100-query-3x__     | 39.4 | 61.8 | 41.9 | 19.7 | 42.9 | 59.3 |   106.1   |   45.5    |

---

> > ### Author Response · Authors · 2022-08-09
> > **Thanks for your review.**
> >
> > Thanks for your review, which helps us clarify our contributions and performance details. Specifically, we add more detailed comparisons between our work and TrackFormer and EfficientVIS to the related work and experiment sections to elucidate our contributions. We also add a table in the supplementary material to display the performance differences between the our base model and QueryInst in terms of COCO instance segmentation.
> >
> > As it is the last day of the Reviewer-Author Discussion session, we would like to know if we have solved your concerns and if there are any other concerns we can address for you. Please feel free to let us know.

---

> > > ### Comment · Reviewer_g8FE · 2022-08-10
> > > **Thanks for your response**
> > >
> > > Thanks for your response and I'm feeling sorry for the delayed reply. The revised version of InsPro covers most of my concerns. The latest revised version is good to me and I decide to upgrade my rating.

---

> > > > ### Author Response · Authors · 2022-08-10
> > > > **Wow!**
> > > >
> > > > Wow! It is very glad to hear that we address most of your concerns. Thank you very much for this kind rating upgrade. We are really delighted to hear this good news. Have a nice one!

---

### Official Review · Reviewer_kcVg · 2022-07-11

**Rating:** 6
**Confidence:** 3
**Soundness:** 3 good
**Presentation:** 3 good
**Contribution:** 3 good

**Summary:**

The paper proposes a framework for the video instance segmentation task. In order to avoid explicit instance association, which is responsible for computation overhead increment, the author designed both instance query and proposal propagation mechanism.
The framework consists of an intra-query attention module, temporally consistent matching, and an additional loss for removing duplicated predictions.

**Questions:**

1. I notice that some of the numbers in Table 1a are different from those in other papers.

For example, according to Table2 in IFC[10] paper, the number of VisTR and IFC are:

Name | AP | AP50 | AP75 | AR1 | AR10

VisTR | 35.6 | 56.8 | 37.0 | 35.2 | 40.2
IFC | 41.2 | 65.1 | 44.6 | 42.3 | 49.6

Although that does not necessarily raise a concern about the result, I wonder what could cause the difference.


2. In L301-313, the author argues the performance difference results from temporal propagation, yet it could simply be caused by the model size difference. For a fair comparison, the author should also provide the model parameter size after the replacement.

**Limitations:**

Please refer to Questions.

**Strengths And Weaknesses:**

[Strengths]

1. The proposed method is simple and reasonable.
2. The paper is clearly presented and well-organized.
3. The proposed method outperforms all baselines including current state-of-the-art methods.

[Weaknesses]
Please refer to Questions.

---

> ### Author Response · Authors · 2022-07-29
> **Rebuttal**
>
> We thank the reviewer for the positive feedback and helpful comments. We address the questions below.
> #### __Questions__:
> __Q1__: Why are some numbers in Table 1 (a) different from those in other papers? \
> __A1__: This is because the experimental results of VisTR used in our paper are taken from arXiv:2011.14503v2, and IFC's from arXiv:2106.03299v1, which seem outdated. Thanks for the reviewer's reminder. We have found that both VisTR and IFC have updated their experimental results in their camera-ready version, which still are inferior to ours. We will check and update the experimental results of all methods in Table 1 in the revised version.
>
> __Q2__: For a fair comparison, the author should also provide the model parameter size after the replacement. \
> __A2__: Thanks for this suggestion. The parameter size and FLOPs comparisons are shown in the table below. Both methods use ResNet-50 as backbone and have an input resolution of 640 x 360. \
> As shown in the table, our InsPro surpasses the 'track-by-detect' model by a large margin even though InsPro has fewer parameters and FLOPs. We think this is because 'track-by-detect' model requires an extra tracking head for explicit instance association, which increases the model size.
>
> Method          |    AP    |   AP50   |   AP75   |   FPS    | Param (M) | FLOPs (G)|
> ---             |:--------:|:--------:|:--------:|:--------:|:---------:|:--------:|
> Track-by-detect |   31.5   |   49.3   |   34.1   |   25.4   |   119.9   |   48.3 |
> Ours            | __37.4__ | __57.6__ | __41.1__ | __26.3__ | __106.1__ | __45.5__|

---

> > ### Author Response · Authors · 2022-08-09
> > **Thanks for your review.**
> >
> > Thanks for your review, which helps us spot and resolve the ambiguities. We hope that we have addressed your concerns in this Rebuttal.
> >
> > As it is the last day of the Reviewer-Author Discussion session, if you have any other questions, please feel free to let us know.

---

> > > ### Comment · Reviewer_kcVg · 2022-08-09
> > > **Reviewer Response**
> > >
> > > The author has addressed my concerns. The rating is updated to weak accept.

---

> > > > ### Author Response · Authors · 2022-08-09
> > > > **It's a nice day.**
> > > >
> > > > Many thanks for raising the rating, which is really inspiring and joyful!

---

### Author Response · Authors · 2022-08-02
**Common Concern**

__Q__: Relation and difference between InsPro and other query propagation methods (TransTrack, TrackFormer, MOTR, EfficientVIS). \
__A__: We first discuss the relation and difference between our InsPro and other query propagation methods to clarify the contributions of our work. As reviewers mentioned, the method that associates objects across frames through query propagation mechanism has been recently explored in several works, such as TransTrack (arXiv2012.15460), TrackFormer (CVPR2022), MOTR (ECCV2022) and EfficientVIS (CVPR2022). This shows the effectiveness and potential of such a new object linking approach. On the other hand, it indicates that there still are many challenges to be solved to make this approach work well. As we see, these challenges include: 1) steadily binding one evolving object query or object query-proposal pair to one specific object across frames, 2) accurately detecting and tracking new objects, 3) effectively suppressing duplicate detections or tracklets, and 4) elegantly handling tough scenarios like occlusion. In what follows, we elaborate on the similarity and difference between our InsPro and existing query propagation methods in terms of these four aspects.

1) Stably binding query and object is the key to the success of 'track-by-query_propagation' mechanism. Our InsPro and other query propagation methods share a similar inter-frame query-object binding mechanism which is realized by a temporally consistent groundtruth-prediction matching in the training process. Here we also want to point out that TransTrack is basically a 'track-by-detect' method, although it propagates object queries. This is because it still needs to explicitly match detection boxes to tracking boxes in each frame, unable to perform object association implicitly.

2) Detecting new objects is a big challenge of query-propagation-based tracking methods.
TransTrack uses a complete new object query set to detect seen and new objects, which is redundant since seen objects can be detected by previous queries. Instead, TrackFormer uses a track query subset selected from the previous frame to detect seen objects and a new object query subset to predict new objects in the current frame. MOTR takes a similar approach to detecting new objects. However, these approaches rely on heuristic rules to build the track query subset, which is not elegant and may harm performance. For example, some track queries with low prediction scores in the previous frame would be removed and are not passed to the current frame. However, these track queries may represent objects with heavy occlusion whose trajectories would break due to the removal. As for EfficientVIS, it does not consider this detecting new objects problem, and its performance will probably be greatly impacted if there are new objects in the next clip.
In contrast, our InsPro simply propagates all object queries produced in the previous frame to the current frame, which is much simpler and more elegant. Thanks to our proposed Box Deduplication Loss, those unmatched queries that are filtered out in the aforementioned methods are pushed away from matched queries and serve as candidate queries in our method, which can be used to detect new objects (see Figure 5 in the supplementary).

3) Duplicate detections or tracklets are a common problem in query-propagation-based tracking methods. TransTrack relies on a high score threshold to keep fewer track queries to alleviate this problem. Similarly, TrackFormer employs NMS to remove duplicate predictions. MOTR builds a temporal aggregation network to learn more discriminative features to address this problem, while EfficientVIS does not discuss this problem.
By contrast, we design a Box Deduplication Loss to suppress duplicates and an Inter-query attention module to enhance queries with their predecessors. Our solution avoids heuristic rules and post-processing steps, and is more effective according to the experimental results (Please refer to the below __Experimental Results of Common Concern__).

4) Tough scenarios for tracking include occlusion and motion blur etc. TransTrack, TrackFormer and MOTR use heuristically selected track queries to associate objects across frames. This may miss objects with heavy occlusion. In comparison, our InsPro keeps all object queries and does not have this concern. Furthermore, we enhance current object queries with their historical ones, which can significantly improve performance in tough scenarios.

Overall, we argue that details make the difference. Although those existing methods also take a query propagation approach to object instance association, our method does better at details. Through analyses and experiments (Please refer to the below __Experimental Results of Common Concern__), we have shown that our method is simpler, more elegant and more effective, and we believe it can provide value to the community.

---

> ### Author Response · Authors · 2022-08-02
> **Experimental Results of Common Concern**
>
> To show the superiority of our method, we conduct more experiments to verify this.
>
> We select the most recent MOTR as a representative method. For a fair comparison, we implement the core query propagation module used in MOTR on the same baseline with us (refer to MOTR's code https://github.com/megvii-research/MOTR). We follow MOTR exactly to set up other model and experiment settings. To exclude the influence of other factors, we do not use temporal feature aggregation in both methods. The tables below show the comparisons on YouTube-VIS 2019 and ImageNet VID. It can be seen that our InsPro outperforms MOTR in both benchmarks.
>
> method          |  AP  | AP50 | AP75 |
> ---             |:----:|:----:|:----:|
> MOTR-YTVIS19    | 37.4 | 56.9 | 40.3 |
> InsPro-YTVIS19  | 38.4 | 57.7 | 41.6 |
>
> method      |  AP  | AP50 | AP75 |
> ---         |:----:|:----:|:----:|
> MOTR-VID    | 38.0 | 57.1 | 39.7 |
> InsPro-VID  | 39.5 | 57.2 | 42.7 |
>
> For further comparison, we also implement an online version of EfficientVIS by simply setting clip length T=1. For a fair comparison, we do not use intra-query attention in our InsPro.
> The tables below list the comparison results on YouTube-VIS 2019 and ImageNet VID respectively. Our InsPro surpasses online EfficientVIS by a large margin.
>
> method                      |  AP  | AP50 | AP75 |
> ---                         |:----:|:----:|:----:|
> online EfficientVIS-YTVIS19 | 36.6 | 55.5 | 40.3 |
> InsPro-YTVIS19              | 38.4 | 57.7 | 41.6 |
>
> method                  |  AP  | AP50 | AP75 |
> ---                     |:----:|:----:|:----:|
> online EfficientVIS-VID | 33.2 | 48.8 | 35.5 |
> InsPro-VID              | 39.5 | 57.2 | 42.7 |

---

### Author Response · Authors · 2022-08-08
**The revised paper version is available.**

Hi all, we have completed and uploaded a revised version of our paper according to your helpful comments. We also updated the supplementary material accordingly. Please check them out. We managed to include most of the responses to your concerns in the revision (most of them can be found in the colored text of the paper). However, due to the 9-page limit of the revised paper, we couldn’t elaborate on every response. We will add more detailed responses in the camera-ready version (camera-ready version can have 10 pages) if our paper is accepted. Thanks.

---

### Meta-Review · Area_Chair_MNir · 2022-08-25

**Recommendation:** Accept
**Confidence:** Certain

**Metareview:**

The paper discusses a method for online video instance segmentation. Reviewers appreciated the proposed method but raised concerns regarding difference of reported results to other papers, method being similar to prior work, and limited novelty. The rebuttal addressed most of the concerns prompting reviewers to increase their rating to an accept recommendation. AC doesn't see reasons to overturn an unanimous reviewer recommendation.

**Award:**

No

---

### Decision · Program_Chairs · 2022-09-14

Accept